# Annual Excess Crude Mortality in Europe during the COVID-19 Pandemic: A Longitudinal Joinpoint Regression Analysis of Historical Trends from 2000 to 2021

Alessandro Rovetta 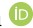

R&C Research, 25073 Bovezzo, Italy; rovetta.mresearch@gmail.com; Tel.: +39-3927112808

**Abstract:** COVID-19 represents the greatest health crisis in recent human history. To date, it is still difficult to estimate its impact on mortality. This paper investigates the excess crude mortality in 27 European countries. The differences between the values observed in 2020 and 2021 with those predicted by a joinpoint regression model were evaluated. A multi-regression analysis was implemented to assess the relationship between health variables and excess mortality. Europe experienced a marked and surprising (S-value > 52) increase in crude mortality during 2020 ($\Delta_\% = +10.0\%$, 95% CI: [2.5; 18.7]) and 2021 ($\Delta_\% = +12.1\%$, 95% CI: [4.3; 21.2]). The difference between average excesses of Eastern and Western countries was not surprising (S < 2) and had little relevance ($\Delta_{E-W} = -2.4$, 95% CI: [−2; 7]) during 2020 but was more pronounced (S = 15, $\Delta_{E-W} = +17.2$, 95% CI: [11.0; 23.5]) during 2021. Excess crude mortality increased in 2021 ($\Delta_\% = +65\%$, 95% CI: [12.6; 118], S = 5.9). Evidence has been found for a surprising and marked negative linear relationship between COVID-19 vaccinations and excess mortality ("*2021 excess mortality = A + BX_4*", with "A = 58 ± 7, S = 28" and "B = −0.65 ± 0.10, S = 22, $R_{adj}^2 = 0.65$, 95% CI: [0.38; 0.82]"). In light of the current literature, these findings provide solid evidence of the substantial role of COVID-19 in the unexpected and marked excess mortality recorded in Europe. COVID-19 vaccinations have appeared to be one of the main determinants for reducing mortality. Future research should explore these aspects in more detail.

**Keywords:** COVID-19; excess mortality; epidemiology; Europe; public health

## 1. Introduction

### 1.1. Background

The COVID-19 pandemic represents the largest health crisis in recent human history. To date, it has caused over 6.5 million official deaths and put health systems worldwide under severe strain [1]. However, various analyses of excess deaths provide evidence of substantial underestimation of global mortality, although the dimension of this effect has varied considerably from study to study [2,3]. In this regard, the percentage of excess deaths directly due to the disease, secondary causes (e.g., overload of health facilities), or tertiary causes (e.g., antiepidemic countermeasures) remains to be clarified. Moreover, the course of COVID-19 has been affected by a wide variety of comorbidities (including cancer, obesity and other disabilities, chronic kidney disease, chronic liver disease, chronic lung diseases, cystic fibrosis, dementia or other neurological conditions, diabetes, heart conditions, HIV infections, and immunocompromised or weakened immune system conditions) and local risk factors (including older age, various types of pollutants, weather and environmental factors, demographic and geographic characteristics, and lifestyles) [4–9]. All this makes it more complex to determine precisely not only the number of deaths due to this novel coronavirus but also the epidemiological risk and the effectiveness of health countermeasures in different countries. Indeed, assessing mortality excesses means assuming a counterfactual scenario—i.e., what would have happened without an anomalous

event—and comparing it with actual observations. This task conceals several pitfalls: it is not just a matter of appropriately choosing and using the measures and tests suitable for the purpose but also finding a model that makes reliable predictions based on historical data [10,11]. Nonetheless, many preliminary analyses of COVID-19 have been limited to comparing the averages of a few years before the pandemic with mortality rates from 2020 onwards [11,12]. Other studies have instead adopted more sophisticated and complex models, which, despite ideally providing more precise results, increase the probability of human errors related to interpreting the outcomes and verifying known and hidden assumptions [10–15]. For instance, a simple comparison of standardized rates assumes a counterfactual scenario with similar populations without considering that the demographic dimension is an important epidemiological determinant whose variation could greatly change the actual outcome (e.g., by altering the relative health care capacity, epidemic diffusion rapidity, social dynamics, or pollution production). For these reasons, not only is it highly improper to compare situations in different countries without modeling these factors, but it is also challenging to find reliable models. Moreover, all this highlights the necessity to evaluate the degree of agreement between different approaches (e.g., simple and complex) in order to have solid evidence. Obviously, the research objective must be weighed against the method adopted (e.g., generally, the danger of an epidemiological event for specific age groups cannot be assessed using only data relating to the total population). In this regard, this paper uses a simple approach: the historical mortality trends in Europe are modeled through a joinpoint linear regression, and the crude mortalities of 2020 and 2021 are compared with the predictions of the model [16,17]. Indeed, the principle of parsimony dictates that once a simple trend (i.e., linear) has been identified, the hypothesis of its continuation is the best option.

*1.2. Research Objectives*

The present study has several research purposes. The first is to provide a general picture of the epidemiological situation in Europe during the COVID-19 pandemic in order to furnish the means to assess the actual public health risk. The second is to investigate the absolute impact of epidemiologically relevant variables such as percentage healthcare expenditure, physicians per 100,000 inhabitants, curative care beds in hospitals per 100,000 inhabitants, and percentage of COVID-19 vaccines. Third, this study is intended to be a guide on the different use and interpretation of statistical surprise and effect size measures. Fourth, this study is intended to be a guide on the adoption and interpretation of statistical procedures concerning counterfactual scenarios. Fifth and last, the analysis is designed to be a standard of transparency and easy reproducibility.

## 2. Materials and Methods

*2.1. Population*

The total inhabitants of the 27 countries of the European Union were considered in this study.

*2.2. Data Collection*

The European crude mortality rates used were those collected by "The World Bank" and "Eurostat" websites [18,19]. The data used regarding the percentage of healthcare expenditure were collected by the "Eurostat" website [20]. The data used regarding the physicians per 100,000 inhabitants were collected by the "Eurostat" website [21]. The data used relating to the curative care beds in hospitals per 100,000 inhabitants were collected by the "Eurostat" website [22]. The data used regarding the percentage of COVID-19 vaccines administered were collected by the "Institute for Health Metrics and Evaluation" website [23].

### 2.3. Study Design

This is a retrospective longitudinal observational study. The raw mortalities of the various European countries were modeled from 2000 to 2019 through a joinpoint linear regression. Outliers were searched for in the distribution of residuals of the latest linear trend identified by the joinpoint regression, including those calculated as differences between the predicted values for 2020 and 2021 and the observed values for 2020 and 2021. Differences between mortality patterns of European countries were sought through comparison tests. Accordingly, it was possible to highlight any anomalies during the COVID-19 pandemic. Through a multi-regression model, any effects of the health variables listed above on the excess crude mortality were evaluated.

### 2.4. Statistical Analysis

Joinpoint and ordinary least squares linear regressions were used to verify the normality conditions of the residuals, homoskedasticity, and absence of multicollinearity [16,17,24,25]. Specifically, joinpoint regression adopts an algorithm that divides the time series into various linear subtrends, thus decreasing the authors' bias on the choice of trends. Parametric and nonparametric tests were selected based on the distributive normality of the samples and exploiting the central limit theorem for datasets with more than 30 elements. The effect size was evaluated using the best values and the associated 95% confidence intervals or standard errors. The statistical surprise was evaluated through the surprisals "S-values = $-\log_2(p\text{-value})$." All the calculations and details are given in the Supplementary Material. Each result was assigned an identification code of the following form "R [number]" (e.g., R1). As a result, the reader can easily associate the tests reported in full in the Supplementary file with the result reported in the paper.

### 3. Results

R1: Overall, as visible in Figure 1, Europe experienced a marked and extremely surprising (S > 52) percentage increase in crude mortality during 2020 ($\Delta_\% = +10.0\%$, 95% CI: [2.5; 18.7]) and 2021 ($\Delta_\% = +12.1\%$, 95% CI: [4.3; 21.2]). The details of the national data are reported in Table 1. R2: The models' residuals were markedly and very surprisingly different from 0, even considering the normalized distribution of the individual countries (average $\Delta = 0.7/1.0$, 95% CI: [0.6; + inf], S > 52). Ergo, there is solid evidence of a widespread and severe crude mortality anomaly recorded throughout Europe during the COVID-19 pandemic. R3: A surprising (S = 5.9) and relevant percentage increase ($\Delta_\% = +65\%$, 95% CI: [12.6; 118]) in mean excess crude mortality was detected between 2020 and 2021.

R4: As shown in Figure 2, there was a discrepancy in the distributions of excess crude mortality between 2020 and 2021. Specifically, in 2021, the Eastern European regions had the highest peaks, while in 2020, the scenario was more homogeneous. Indeed, the difference between average excesses of East (E) and West (W) was not surprising (S < 2) and had little relevance ($\Delta_{E-W} = 2.4$, 95% CI: [−2; 7]) during 2020, however, it was more evident (S = 15, $\Delta_{E-W} = +17.2$, 95% CI: [11.0; 23.5]) during 2021 (see the extremely distant 95% confidence intervals to evaluate the direct comparison).

R5: The multi-linear regression model "2021 excess crude mortality = f (**X**)", with "**X**" equals to the vector "[$X_1$, $X_2$, $X_3$, $X_4$, $X_5$] = [percentage healthcare expenditure, physicians per 100,000 inhabitants, curative care beds in hospitals per 100,000 inhabitants, percentage of COVID-19 vaccines (1+ doses), percentage of COVID-19 vaccine first boosters]," gave the best result "2021 excess mortality = A + B$X_4$," with "A = 58 ± 6, S = 28" and "B = −0.65 ± 0.10, S = 22, $R_{adj}^2 = 0.65$, 95% CI: [0.38; 0.82]" providing evidence of the predominant effects of COVID-19 vaccinations compared to other relevant health variables (Figure 3). On the contrary, the model "2020 excess crude mortality = g (**X**)", with "**X**" equals to the vector "[$X_1$, $X_2$, $X_3$]," gave uncertain results for the best model "2020 excess crude mortality = C + D$X_3$," with "C = 1.4 ± 3.6, S < 2" and "D = 0.017 ± 0.009, S = 3.7, $R_{adj}^2 = 0.08$, 95% CI: [−0.01, 0.36]" (the negative sign has been improperly reported to show

the extension of the confidence interval of $R_{adj}$). Finally, the model "2019 crude mortality = h (**X**)", with "**X**" equals to the vector "[$X_1$, $X_2$, $X_3$]," gave the best model "2019 crude mortality = E + $FX_3$," with "E = 6.5 ± 1.4, S = 13" and "F = 0.010 ± 0.004, S = 7.3, $R_{adj}^2$ = 0.23, 95% CI: [0.02, 0.53]." R6: The percentage change in excess mortality from 2020 to 2021 had a median close to 0 (m = −7, IQR: [−83; 69]) in Western European countries and far from 0 (m = 152, IQR: [88; 439]) in Eastern European countries. The difference between the two distributions was surprising (S = 7.6). R7: Changes in excess mortality were moderately correlated with COVID-19 vaccination rates (R = −0.37, 95% CI: [−0.64; −0.03], S = 4.2).

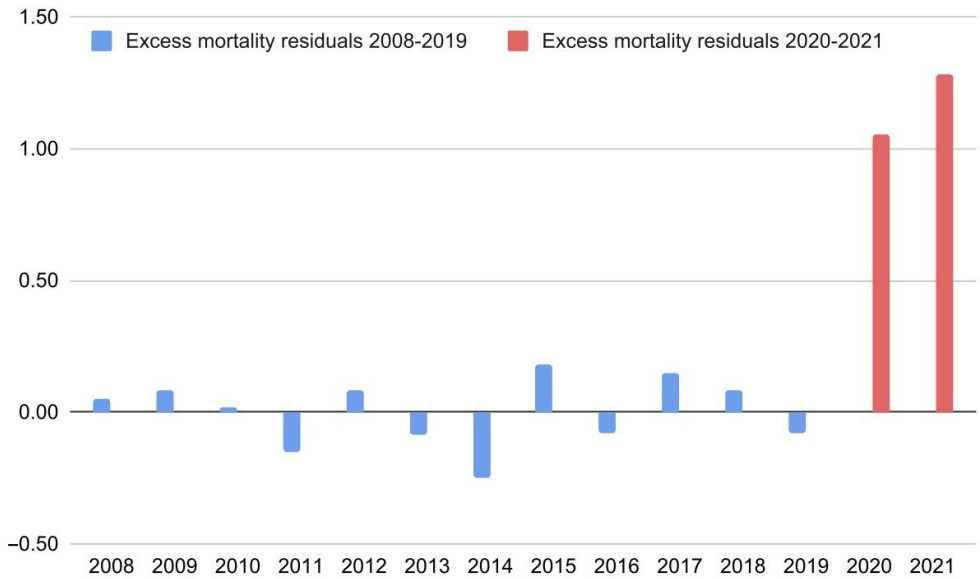

**Figure 1.** Excess mortality residuals (differences between the model's predictions and observed data) in Europe from 2008 to 2021. The data shown concern the latest linear trend identified by the joinpoint regression (2008–2019).

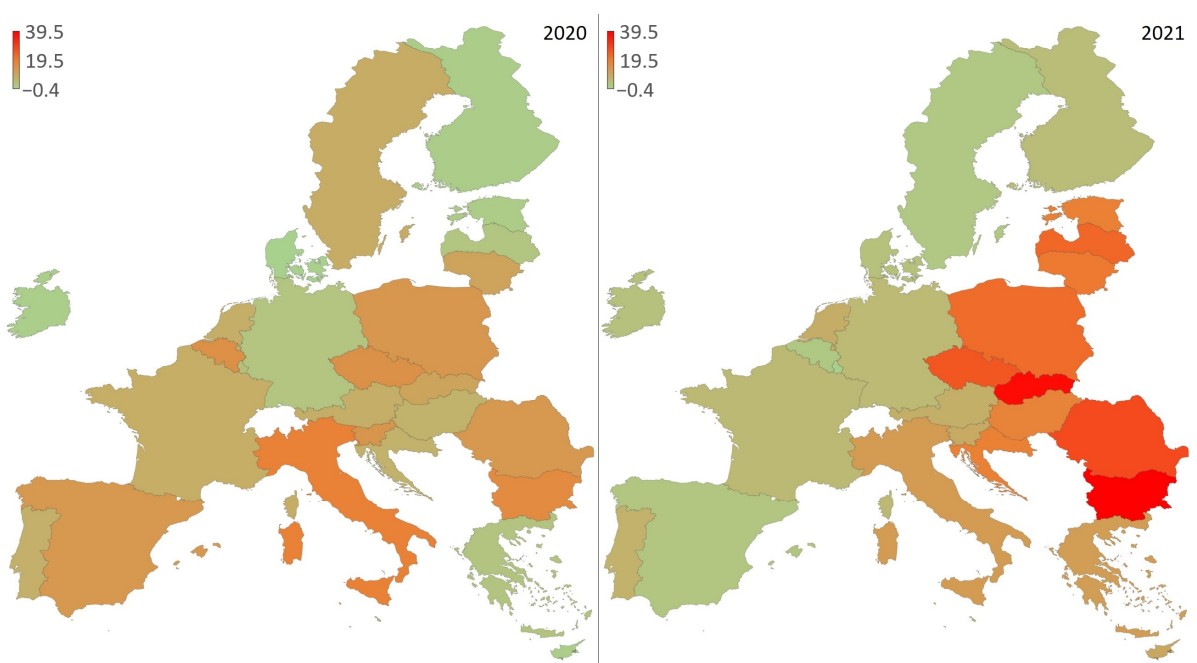

**Figure 2.** Differences in excess crude mortality in the 27 countries of the European Union during 2020 and 2021. The values shown correspond to the percentage increases (compared to the predictions of the joinpoint regression).

**Table 1.** Differences between expected and observed crude mortalities in the 27 countries of the European Union in 2020 and 2021.

| Country | Predicted 2020 | Observed | Δ% 2020 | Predicted 2021 | Observed | Δ% 2021 |
|---|---|---|---|---|---|---|
| Austria | 9.5 | 10.3 | 7.9 | 9.6 | 10.3 | 7.5 |
| Belgium | 9.6 | 11 | 14.8 | 9.6 | 9.7 | 1.3 |
| Bulgaria | 15.5 | 18 | 16.2 | 15.6 | 21.7 | 39.5 |
| Croatia | 13.2 | 14.1 | 7.0 | 13.3 | 15.8 | 18.5 |
| Cyprus | 7.2 | 7.2 | 0.4 | 7.2 | 7.9 | 9.1 |
| Czech Republic | 10.6 | 12.1 | 14.7 | 10.6 | 13.3 | 25.6 |
| Denmark | 9.4 | 9.4 | −0.4 | 9.5 | 9.8 | 3.2 |
| Estonia | 11.8 | 11.9 | 0.9 | 11.8 | 14 | 18.5 |
| Finland | 9.9 | 10 | 0.6 | 10.0 | 10.4 | 4.1 |
| France | 9.2 | 9.9 | 8.0 | 9.2 | 9.7 | 5.1 |
| Germany | 11.6 | 11.9 | 2.5 | 11.7 | 12.3 | 4.9 |
| Greece | 11.9 | 12.2 | 2.6 | 12.1 | 13.5 | 11.6 |
| Hungary | 13.6 | 14.5 | 6.8 | 13.7 | 16.1 | 17.8 |
| Ireland | 6.4 | 6.4 | 0.2 | 6.4 | 6.6 | 3.1 |
| Italy | 10.6 | 12.6 | 18.5 | 10.7 | 12 | 12.2 |
| Latvia | 14.9 | 15.2 | 2.2 | 15.0 | 18.4 | 23.0 |
| Lithuania | 14.1 | 15.6 | 10.4 | 14.2 | 17 | 20.0 |
| Luxembourg | 7.0 | 7.3 | 3.9 | 7.0 | 7 | -0.6 |
| Malta | 7.6 | 7.9 | 4.5 | 7.6 | 8 | 5.9 |
| Netherlands | 9.0 | 9.7 | 7.8 | 9.1 | 9.8 | 7.9 |
| Poland | 11.1 | 12.6 | 13.4 | 11.3 | 13.8 | 22.2 |
| Portugal | 11.2 | 12 | 7.4 | 11.3 | 12.1 | 6.8 |
| Romania | 13.6 | 15.4 | 13.2 | 13.7 | 17.5 | 27.6 |
| Slovak Republic | 9.8 | 10.8 | 10.3 | 9.8 | 13.5 | 37.8 |
| Slovenia | 10.0 | 11.4 | 13.5 | 10.1 | 11 | 8.4 |
| Spain | 9.2 | 10.4 | 13.0 | 9.3 | 9.5 | 2.2 |
| Sweden | 8.8 | 9.5 | 8.4 | 8.7 | 8.8 | 1.5 |
| EUROPE | 10.5 | 11.6 | 10.0 | 10.6 | 11.9 | 12.1 |

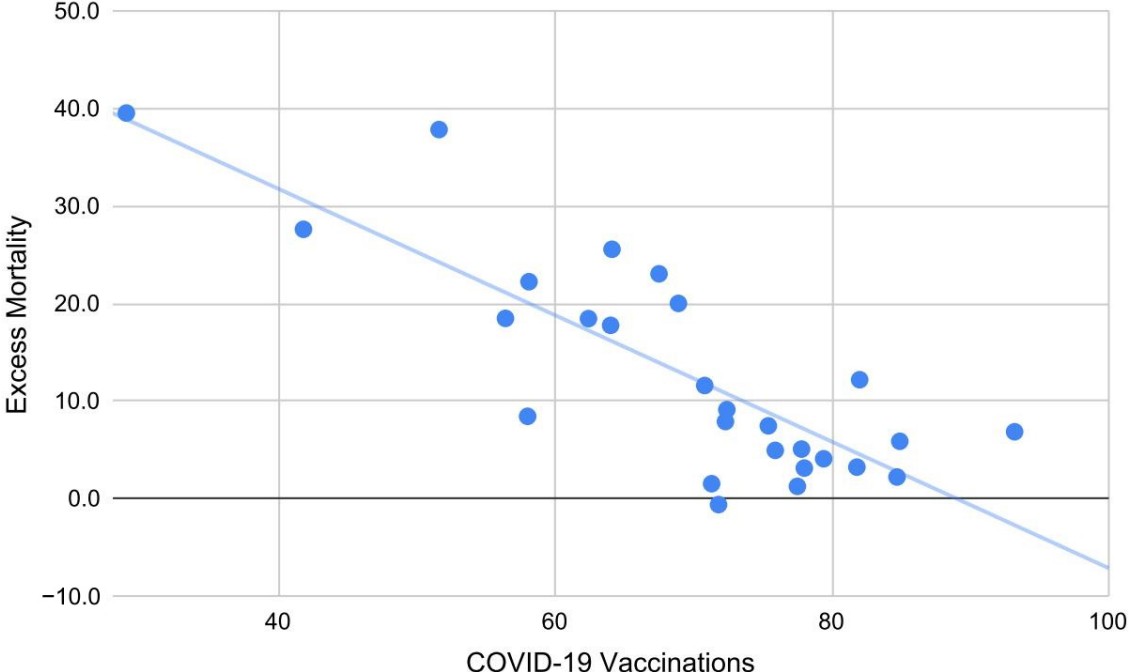

**Figure 3.** Excess mortality in European countries during 2021 as a linear function of COVID-19 vaccines (1 + doses). Each additional percentage point in vaccinations corresponds to a decrease in mortality of about 0.65 ($R_{adj}^2 = 0.65$).

## 4. Discussion

### 4.1. Principal Findings

The counterfactual analysis based on joinpoint linear regression predictions provided evidence of a widespread, drastic, and unexpected increase in crude mortality in Europe during the COVID-19 pandemic. These findings support the underestimation of the COVID-19 impact on mortality in Europe. No specific patterns were identified in 2020, while in 2021, Eastern European countries experienced a statistically surprising greater increase in crude mortality. In general, excess crude mortality during 2021 was higher than in 2020, although, as visible in Figure 2, many Western European countries have witnessed a slight decline. Linear multi-regression models have provided evidence in favor of a marked and statistically surprising negative correlation between excess crude mortality and COVID-19 vaccinations during 2021 and a moderate positive correlation between excess crude mortality and care beds rate in hospitals during 2020 and, especially, in 2019. Since the mortality reduction in COVID-19 patients due to vaccines has been scientifically demonstrated, the first data support the hypothesis that the most anomalous deaths increase was directly caused by SARS-CoV-2 infection. Furthermore, COVID-19 vaccinations appear to be a major epidemiological determinant for decreasing crude mortality. On the contrary, the positive correlation between care beds and mortality (present even before the pandemic) plausibly signals the higher concentration of care beds where epidemiological risks are more pronounced, i.e., a public health strategy.

### 4.2. Comparison with Other Literature

In general, these findings on excess mortality are in agreement with the previous literature, although the effect size and significance has varied from study to study [3,13,26]. Specifically, the excess crude mortality clearly exceeded that certified by COVID-19 swabs during 2020 and 2021. While it cannot be assumed that all excess mortality is directly due to SARS-CoV-2 infection, it must be considered that the ability of a viral agent to compromise the health system and act as a risk factor for other diseases is an intrinsic part of its epidemiological danger [11]. Indeed, characteristics such as mortality (deaths over the exposed population) and lethality (deaths over cases of infection) cannot be abstracted, as they depend on the epidemiological context in which the virus spreads [4–9]. In general, the increase in excess crude mortality during 2021 is compatible with the circulation of variants such as alpha, beta, gamma, and delta, which have had greater transmissibility and virulence [27]. As far as vaccines are concerned, since an unprecedented amount of evidence is provided in the literature on their shielding effect against the lethality of COVID-19 [28–34], the negative relationship found between the percentage of doses and excess crude mortality provides evidence of a decisive effect of COVID-19 vaccines in mitigating general mortality in Europe. Furthermore, this also provides evidence that the excess crude mortality is mainly due to COVID-19. The exclusion from the regression model of variables such as percentage healthcare expenditure, number of physicians per inhabitant, and curative care beds in hospitals per inhabitant does not imply their uselessness but simply that (i) the effects could have been masked by confounding factors or local inhomogeneities, and (ii) vaccination coverage may have had a predominant role in reducing excess mortality. For instance, beyond mere health care, physicians play a vital role in disseminating vaccine information among the population [35]. Furthermore, the lack of healthcare facilities, beds, instruments, and personnel was one of the central problems of COVID-19 management [36,37]. Indeed, this crisis has prompted the scientific community to demand that the capacity and funding of health systems be more flexible in dealing with exceptional emergencies and that human resource recruitment be planned and financed with a long-term vision.

### 4.3. Practical Implications

Based on the above findings and considerations, the following public health strategies are recommended. First, linear joinpoint regression, being little exposed to author bias and

quick to use, should be adopted as a surveillance method to highlight statistical anomalies related to novel epidemiological events. In particular, comparing observed and predicted data provide relevant information on the actual epidemiological risk and situation. Second, the determinants of the difference in mortality between Eastern and Western Europe need to be explored further so that targeted improvement interventions can be made. Third, based on the previous point, the European Community is called to intervene to help the countries of Eastern Europe not only concerning the present health crisis but also to reduce the epidemiological risk in general. Four, we recommend using these findings to evaluate the overall impact of the COVID-19 vaccination campaign. In this regard, the negative correlation between mortality and vaccination rates suggests that more vaccinations would have been beneficial, especially in Eastern countries. Five, since the mortality of COVID-19 has been underreported, it is imperative that states invest in more efficient and effective testing systems in anticipation of other new epidemics. Finally, as excess mortality was high and uniform across Europe in 2020, more funds should be allocated to national healthcare systems to prevent a similar situation from happening again in the future.

*4.4. Limitations*

This research has limitations to be considered. The evidence from this study will not allow a direct comparison of the anti-COVID-19 policies of the various European countries since too many of the known confounding factors have not been adequately modeled. Despite being underlined by the authors of this kind of research [38], this aspect is often responsible and dangerously omitted or ignored by communicators [39]. Moreover, causal evidence cannot be inferred from these results alone. For instance, this paper is not designed to determine the efficacy of COVID-19 vaccines, but only to analyze the scenario in light of the evidence reported in the previous literature. Finally, no comparisons between single countries have been performed (these would have required specific tests since multiple single tests do not allow direct comparisons).

*4.5. Conclusions*

These findings provide solid evidence of a substantial and surprising increase in crude mortality across Europe during the COVID-19 pandemic. In 2021, Eastern countries experienced a greater increase in excess crude mortality compared to that experienced by Western countries, while the situation was more homogeneous in 2020. These aspects are of fundamental importance for assessing the global health emergency in various countries. In light of what is already known in the literature, the negative linear relationship between COVID-19 vaccines and excess crude mortality provides evidence in favor of the major relevance of the vaccination campaign as a strategy for safeguarding public health compared to other factors related to medical assistance. Moreover, this result provides evidence that excess crude mortality is mostly due to the COVID-19 pandemic as a direct consequence of the SARS-CoV-2 infection. Future research should explore these aspects in more detail by analyzing the impact of possible confounding factors. Furthermore, national health systems should be strengthened in anticipation of future epidemics and crises.

**Supplementary Materials:** The following supporting information can be downloaded at: https://osf.io/5gf6w/ (accessed on 14 November 2022).

**Funding:** This research received no external funding.

**Institutional Review Board Statement:** Not applicable.

**Informed Consent Statement:** Not applicable.

**Data Availability Statement:** All data used or generated in this study are reported in the manuscript, Supplementary Material, or cited references.

**Acknowledgments:** I would like to thank Lucia Castaldo for her support during the writing of this manuscript.

**Conflicts of Interest:** The author declares no conflict of interest.

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
