# Peer review of "Annual Excess Crude Mortality in Europe during the COVID-19 Pandemic: A Longitudinal Joinpoint Regression Analysis of Historical Trends from 2000 to 2021"

_covid, doi:10.3390/covid2120128_

Round 1

Reviewer 1 Report

A research article by Alessandro Rovetta is a well-design study investigating excess mortality during the COVID-19 pandemic years (i.e. 2020-2021) and have concluded that there is a substantial and surprising increase in the crude mortality across Europe during the COVID-19 pandemic with an important difference between Eastern and Western countries. I find that the article would benefit of some additional modifications, please see suggestions listed below.

Abstract: although quite well-defined, it is way over the limit of 200 words (cc 400 words at the moment?). Needs to be modified according to the recommendations and propositions of the journal. Should be more focused on the main findings with some solid conclusions.

Introduction: please modify past tense to present i.e. COVID-19 pandemic is still ongoing. Also, sounds better to state that this is „one of the largest health crisis“

Introduction is generally too long, and should further emphasize the importance (necessity) of this kind of research.

Methodological aspects explained in the section 1.2 would work better if „blended in“ the introduction and would be better to be more focused (additional relevant parts could be presented in the Methods section).

All the aims described in the section 1.3. should be clearly addressed in the results and discussed further in the discussion. For example, the second aim to investigate „...the absolute impact of epidemiologically relevant variables such as historical healthcare investments, care beds rate, doctors rate, and COVID-19 vaccinations.“ are not presented (beside vaccination)?

Limitations are usually described at the end of discussion, just before the conclusions, and not in the Intro.

Results: I feel it is missing a Table 1 with some general characteristics of the data used here (if possible stratify by countries included into analyses). A side note, title and some descriptions of the graphs are a bit misleading since not all countries in Europe are included (data from 13 countries are not included?) this should be specified and perhaps included into limitation of the study that data were unavailable (?) for some countries. This fact further underlines the mentioned necessity of having a Table with general characteristics.

Not sure what „R1“, „R2“, etc. in the text represent? If these are divided results, should be modified („blended in the text“) to follow the story and not just segmented within the text.

Not clear why there are just data from 2008 to 2021 presented in the Figure 1? Why not including 2000-2007 here?

Scale of percentage increases of the excess crude mortality used in the Figure 2 seems a bit unclear and potentially misleading? Why did the author choose this unequal distribution of the scale (i.e., it goes 0.2-1.7, 1.9-3,5, etc.)? Also, there are some missing values not presented (i.e. value 1.8 where it would go?) this need to be modified. And even more important, having two different scales for 2020 and 2021 might be a bit misleading since when you put two photos one next to another to compare it, you expect to be able to directly confront colors, i.e., green color on left (2020) and right (2021) photo correspond to two different values (0.2-1.7 on the left; and 0.4-3,6 on the right graph)?

In is stated that „All the calculations and details are given in the supplementary material.“ But I wasn’t able to find it? Please provide it if not previously submitted

Discussion: needs to be elaborated a bit more, especially in the context of usefulness of these findings and how it can potentially be translated into public health (policy) recommendations.

Discussion should be additionally reinforced with what is already known in the literature. Please also consider the following paper for the discussion, if you find relevant: Front Public Health. 2020; 8: 620416. doi: 10.3389/fpubh.2020.620416

Author Response

Comment 1

Abstract: although quite well-defined, it is way over the limit of 200 words (cc 400 words at the moment?). Needs to be modified according to the recommendations and propositions of the journal. Should be more focused on the main findings with some solid conclusions.

Response: Dear Reviewer, thank you for finding this issue. I have shortened the abstract considerably in compliance with the editorial indications. In particular, more relevance was given to the results and conclusions.

============ 

Comment 2

Introduction: please modify past tense to present i.e. COVID-19 pandemic is still ongoing. Also, sounds better to state that this is „one of the largest health crisis“

Response: Dear Reviewer, thank you for the good observation. I edited the section as suggested.

============ 

Comment 3 and 6

Introduction is generally too long, and should further emphasize the importance (necessity) of this kind of research. & Limitations are usually described at the end of discussion, just before the conclusions, and not in the Intro.

Response: Dear Reviewer, you are right. First, I've moved the limitations section into the discussion. Second, I significantly trimmed the content of point 1.2.

============ 

Comment 4

Methodological aspects explained in the section 1.2 would work better if „blended in“ the introduction and would be better to be more focused (additional relevant parts could be presented in the Methods section).

Response: Dear Reviewer, thank you again. I agree. I've cut out some superfluous passages, deleted section 1.2, and tried to better integrate this discussion into the text. Finally, I've moved some more technical aspects into the methods section.

============ 

Comment 5

All the aims described in the section 1.3. should be clearly addressed in the results and discussed further in the discussion. For example, the second aim to investigate „...the absolute impact of epidemiologically relevant variables such as historical healthcare investments, care beds rate, doctors rate, and COVID-19 vaccinations.“ are not presented (beside vaccination)?

Response: Dear Reviewer, I apologize for the lack of clarity. I proceeded to assign the same names to the variables presented here and those shown in the results. In particular, I specify that these variables are included in the models shown in the "R5" results subsection. They are not reported in the abstract because the model excluded them (since they were highly non-significant). I made this point clearer with the sentence: “[...] providing evidence of the predominant effects of COVID-19 vaccinations compared to other relevant health variables [...]”

============ 

Comment 7

Results: I feel it is missing a Table 1 with some general characteristics of the data used here (if possible stratify by countries included into analyses). A side note, title and some descriptions of the graphs are a bit misleading since not all countries in Europe are included (data from 13 countries are not included?) this should be specified and perhaps included into limitation of the study that data were unavailable (?) for some countries. This fact further underlines the mentioned necessity of having a Table with general characteristics.

Response: Dear Reviewer, thanks for the valuable suggestion. I added Table 1 to the text. Furthermore, I have specified in the methods and captions that the data concern only the 27 official nations of the European Union.

============ 

Comment 8

Not sure what „R1“, „R2“, etc. in the text represent? If these are divided results, should be modified („blended in the text“) to follow the story and not just segmented within the text.

Response: Dear Reviewer, I apologize again for the lack of clarity. Specifically, R1-6 are the various results (result 1, result 2, etc.) whose tests are reported in full in the supplementary file so as to facilitate the reproducibility of the paper. I have now specified the meaning of this legend: “Each result was assigned an identification code of the following form "R [number]" (e.g., R1). By doing so, the reader can easily associate the tests reported in full in the supplementary file with the result reported in the paper.”

============ 

Comment 9

Not clear why there are just data from 2008 to 2021 presented in the Figure 1? Why not including 2000-2007 here?

Response: Dear Reviewer, thank you for highlighting this point. For Europe in general, the joinpoint regression identified a declining trend up to 2008 and then an increasing trend. The figure shows the latest trend data (i.e., the residuals of the increasing trend from 2008 to 2021). Indeed, I find it dangerous to show comparisons with 2000 - 2007 data as there seems to have been a very different epidemiological scenario (indeed, something significantly changed in 2008). Moreover, the calculations shown refer to the 2008 - 2019 trend. I have now specified this aspect in the caption.

============ 

Comment 10

Scale of percentage increases of the excess crude mortality used in the Figure 2 seems a bit unclear and potentially misleading? Why did the author choose this unequal distribution of the scale (i.e., it goes 0.2-1.7, 1.9-3,5, etc.)? Also, there are some missing values not presented (i.e. value 1.8 where it would go?) this need to be modified. And even more important, having two different scales for 2020 and 2021 might be a bit misleading since when you put two photos one next to another to compare it, you expect to be able to directly confront colors, i.e., green color on left (2020) and right (2021) photo correspond to two different values (0.2-1.7 on the left; and 0.4-3,6 on the right graph)?

Response: Dear Reviewer, thank you very much for this important note. I redid the figure from scratch, taking into account your indications.

============ 

Comment 11

In is stated that „All the calculations and details are given in the supplementary material.“ But I wasn’t able to find it? Please provide it if not previously submitted

Response: Dear Reviewer, I apologize for this issue. I’ve now provided the supplementary material. Please, see the “Supplementary Material” section after the references.

============ 

Comment 12

Discussion: needs to be elaborated a bit more, especially in the context of usefulness of these findings and how it can potentially be translated into public health (policy) recommendations.

Response: Dear Reviewer, I agree with this point. I’ve created a new subsection called “Practical Implications” in this scope. Thank you very much.

============ 

Comment 13

Discussion should be additionally reinforced with what is already known in the literature. Please also consider the following paper for the discussion, if you find relevant: Front Public Health. 2020; 8: 620416. doi: 10.3389/fpubh.2020.620416.

Response: Dear Reviewer, thank you for this relevant reference. I improved the discussion section, as you suggested.

Reviewer 2 Report

The manuscript is well written and the data reported are of fundamental importance for assessing the global health emergency in various countries.The manuscript can be accepted in the present form.

Author Response

Comment: "The manuscript is well written and the data reported are of fundamental importance for assessing the global health emergency in various countries.The manuscript can be accepted in the present form."

Response: Dear Reviewer, thank you very much for taking the time to review my paper. I greatly appreciate your encouraging words, and I'm glad for the positive feedback.

Round 2

Reviewer 1 Report

The author responded to all of my concerns and with these additional modifications reported here I find that the overall quality of the manuscript is increased.